# CT Perfusion in Lacunar Stroke: A Systematic Review

**DOI:** 10.3390/diagnostics13091564

**Published:** 2023-04-27

**Authors:** Marialuisa Zedde, Manuela Napoli, Ilaria Grisendi, Federica Assenza, Claudio Moratti, Franco Valzania, Rosario Pascarella

**Affiliations:** 1Neurology Unit, Stroke Unit, AUSL-IRCCS di Reggio Emilia, via Amendola 2, 42122 Reggio Emilia, Italy; grisendi.ilaria@ausl.re.it (I.G.);; 2Neuroradiology Unit, AUSL-IRCCS di Reggio Emilia, via Amendola 2, 42122 Reggio Emilia, Italy; napoli.manuela@ausl.re.it (M.N.);

**Keywords:** perfusion, CTP, stroke, lacunar stroke, small vessel disease, SVD, MRI, lacuna, RSSI, recent small subcortical infarct

## Abstract

Background. The main theory underlying the use of perfusion imaging in acute ischemic stroke is the presence of a hypoperfused volume of the brain downstream of an occluded artery. Indeed, the main purpose of perfusion imaging is to select patients for endovascular treatment. Computed Tomography Perfusion (CTP) is the more used technique because of its wide availability but lacunar infarcts are theoretically outside the purpose of CTP, and limited data are available about CTP performance in acute stroke patients with lacunar stroke. Methods. We performed a systematic review searching in PubMed and EMBASE for CTP and lacunar stroke with a final selection of 14 papers, which were examined for data extraction and, in particular, CTP technical issues and sensitivity, specificity, PPV, and NPV values. Results. A global cohort of 583 patients with lacunar stroke was identified, with a mean age ranging from 59.8 to 72 years and a female percentage ranging from 32 to 53.1%.CTP was performed with different technologies (16 to 320 rows), different post-processing software, and different maps. Sensitivity ranges from 0 to 62.5%, and specificity from 20 to 100%. Conclusions. CTP does not allow to reasonable exclude lacunar infarct if no perfusion deficit is found, but the pathophysiology of lacunar infarct is more complex than previously thought.

## 1. Introduction

Advanced neuroimaging has been increasingly used in the last years for selecting acute stroke patients for revascularization treatment. Both Computed Tomography (CT)- based and Magnetic Resonance Imaging (MRI)-based protocols have been validated for this purpose and practically applied as suggested by the guidelines, but multimodal CT is becoming the preferred imaging assessment for acute stroke patients [1,2]. Regardless of technique, the two main aims of advanced neuroimaging are the identifications of large vessel occlusion (LVO) or distal vessel occlusion (DVO) amenable for endovascular treatment and the assessment of tissue viability in the corresponding territory, particularly in the late time window. In particular, in the time window between 4.5 and 9 h. the identification of a core/penumbra mismatch is a mandatory requirement to administer rtPA, independently from the demonstration of vessel occlusion. Indeed, the perfusion thresholds were assessed with an automated processing software and mediated from the clinical trials and their meta-analysis [3] (i.e., an infarct core volume < 70 mL and a critically hypoperfused volume/infarct core volume > 1.2 with mismatch volume > 10 mL) allowed rtPA administration in patients for whom mechanical thrombectomy is either not indicated or not planned [4]. Moreover, in patients with unknown time onset of symptoms and wake-up stroke, who have CT or MRI core/perfusion mismatch within 9 h from the midpoint of sleep, and for whom mechanical thrombectomy is either not indicated or not planned, rtPA administration is recommended with a definition of mismatch threshold as previously detailed [5].

The premise of perfusion imaging in acute stroke is the presence of territorial ischemia due to LVO or DVO. Lacunar stroke, expression mainly of small vessel disease or atherothrombotic involvement of the parent artery occluding a perforating branch, is outside this purpose, so there is only scarce information on the use of perfusion techniques in the hyperacute phase for the identification of a mismatch in the territory of a perforating artery. Perfusion changes sized as lacunar infarcts are not visible on the post-processed core-penumbra maps because they are typically smoothed on automated software, including in the map only relatively large clusters of hypoperfused pixels. A few small studies [6,7,8] suggested that visual assessment of the mean transit time (MTT) map on CTP had reasonable sensitivity to detect lacunar infarction. Diffusion Weighted Imaging (DWI) MRI has excellent sensitivity for lacunar stroke, but, unfortunately, it is not largely accessible in the management of hyperacute stroke. CTP has advantages in accessibility, but a couple of studies [9,10] found 50% of false negative cases could be attributed to lacunar stroke.

The aim of this systematic review is to define, according to the available data, the diagnostic yield of CTP in lacunar ischemic stroke in the hyperacute phase.

## 2. Materials and Methods

### 2.1. Sources

This systematic review follows the Meta-Analyses and Systematic Reviews of Observational Studies (MOOSE) group guidelines [11].

We searched PubMed and EMBASE databases for studies addressing CTP findings in patients with acute lacunar stroke published from 1 January 2000 to 31 December 2022. We used the following keywords respectively for PubMed and EMBASE: CTP OR “CT perfusion” AND (lacunar OR lacuna OR lacunar infarct*); (‘ct perfusion’/exp OR ‘ct perfusion’ OR ((‘ct’/exp OR ct) AND (‘perfusion’/exp OR perfusion))) AND (‘lacunar stroke’/exp OR ‘lacunar stroke’). In addition, we applied forward and backward citation tracking to improve the results.

### 2.2. Eligibility Criteria

All studies presenting original data that reported CTP performance in lacunar stroke identification were included. We limited the selection to English-language studies and excluded case reports and studies on nonhuman subjects. We also excluded studies evaluating CTP in acute stroke without explicitly including lacunar stroke and, in particular, studies on LVO and DVO. Abstracts presented at relevant scientific meetings were excluded because of the lack of relevant information. We avoided including duplicated datasets. 

We relied on the definition and diagnostics provided in the original study for lacunar infarct definition and CTP maps and parameters descriptions and for automatic post-processing analysis. No limitation was predefined according to the number of rows of CT scanners.

Two investigators (MZ, RP) independently screened the papers retrieved in the literature search and performed them accordingly to the previously detailed criteria.

### 2.3. Data Extraction

The NIH Quality Assessment Tool for Observational Cohort and Cross-Sectional Studies [12] was used to evaluate each eligible publication. The following information was extracted: authors, year publication, country, study design, population features, CTP technical details, sensitivity, specificity, positive predictive value (PPV), and negative predictive value (NPV) in comparison with follow-up neuroimaging findings. 

In case of missing values, we tried to derive them whenever it is possible [12]. Disagreements between the two reviewers were addressed and resolved by consensus.

## 3. Results

We identified 99 papers, and after the screening, we selected 14 publications (see PRISMA table in Figure 1). 

According to the NIH Quality Assessment Tool for Observational Cohort and Cross-Sectional, all studies achieved a fair level of quality.

The selected studies cover a wide temporal range in enrolling patients from 2004 [9] to 2021 [13], and the main features of inclusion criteria and lacunar stroke definition were summarized in Table 1. Several countries on multiple continents are represented with missing studies/data for African and South American people. All studies were retrospective case series from comprehensive stroke centers. 

Most studies made explicit the definition of lacunar infarction based on neuroimaging (mainly DWI-MRI at the follow-up), but a minority of papers did not provide this information. Only a few studies referred to standardized neuroimaging reporting for SVD stroke [7].

The demographic features of the enrolled patients and the neuroimaging technical details and findings, as extracted from the selected papers, are summarized in Table 2. 

The selected papers provided data on a global sample of 583 patients with lacunar stroke (two papers did not specify the number of lacunar strokes within the study population). The mean age ranges from 59.8 [6] to 72 years [16], with a women percentage ranging from 32 [14] to 53.1% [6], but with the half of studies (7) not reporting the information. In three studies [9,14,17], CTP was performed with a 16 rows scanner; in one study [17], a 40 rows scanner was used; in five studies [6,9,15,16,18], a 128 rows scanner was declared; in two studies [15,21], a 320 rows scanner was used allowing a whole brain perfusion approach and in one study [20] no technical detail was provided. A great variety of post-processing software was reported, ranging from vendor machines tools [9,16,17,18,19,23,24,25] to commercially available tools not related to individual vendor machines, such as Vitrea [21,21], RAPID [14], MIStar [15,23], F-STROKE [13]. However, most studies relied on visual assessment of the CTP maps and not on the automatic identification of hypoperfused areas. A consistent variability was reported in the perfusion maps examined in the individual studies, as depicted in Table 3. Table 3 also reported the vertical coverage of CTP package acquisition as indicated in the individual studies, and Table 4 the sensitivity, specificity, PPV, and NPV values in the selected studies.

## 4. Discussion

The core-penumbra hypothesis has been well documented in acute ischemic stroke due to LVO and DVO, and it is the conceptual framework of perfusion imaging and reperfusion treatment. In lacunar stroke, where the presumed occluded vessel is a single small perforating artery, this hypothesis is still a matter of debate. Looking for RSSI by using CTP is based on the hypothesis that stroke due to small vessel occlusion has a perfusion deficit as stroke due to LVO and DVO. Addressing the role of CTP in RSSI diagnosis is the main aim of our systematic review, and the results are really heterogeneous among the studies. Several sources of variability can be identified, and they substantially belong to two categories: technical or methodological issues about CTP and the pathophysiological heterogeneity of RSSI. In particular, this last point has not been systematically considered, and it may potentially affect the CTP performance. The relevance of RSSI pathophysiology in interpreting the CTP results makes it the first issue to consider. 

About the first issue, the presumed evolution of RSSIs into lacunes has led to the term lacunar infarction based on careful histopathologic examinations, but neuroimaging follow-up [28] provided the information that RSSIs do not always evolve into lacunes but can remain as mainly non-cavitated-white matter hyperintensities (WMH) or even disappear after several weeks or months. The underlying pathophysiology tells us something about the possible differences in tissue vulnerability and repair, and the eventual relation with perfusion status at baseline is still unknown. 

Regarding the second issue, i.e., technical issues at CTP, we are proposing some considerations. The sensitivity of CTP varies considerably due to the heterogeneity in patient characteristics, CTP spatial and temporal resolution, and post-processing methods [29], being higher in atherothrombotic stroke [18]. TTP or Tmax maps are highly sensitive to LVO and collateral blood flow [23]. Some studies about detecting acute lacunar infarcts with CTP found a sensitivity of 48.7–56% and a specificity of 98.7–100% [6,9,14]. Nevertheless, CTP maps are not good enough to evaluate lacunar infarcts due to their lower resolution and may result in false-negative results [29]. CBF and CBV abnormalities were more frequent among patients with Branch Artery Disease (BAD) than in lacunar infarct patients [24], probably because of the larger size of BAD, usually involving more than one perforating artery [30]. 

Sensitivity and specificity data for CTP diagnosis of lacunar stroke, as shown in Table 3, are available from seven studies and four of them provided detailed information. All the available studies were retrospective, and this limitation may affect global findings. There are several other sources of heterogeneity: the definition of lacunar stroke, the site and size of lacunar infarction (or RSSI), the year of patients’ enrollment, the rate of rtPA administration, the DWI-MRI diagnosis and timing of MRI in follow-up, and finally the scanner’s technology (scanner, volume coverage, post-processing software, and visual assessment, perfusion maps).

The definition of lacunar stroke used in the selected studies ranged from the partially overlapping concept of RSSI to the more complicated diameter-based definition (higher threshold of 15 mm [23] to 25 mm [14]) to the combination of clinical lacunar syndromes, hyperintensity on DWI-MRI in a penetrating vessel territory and no visible vessel occlusion in that region [15]. The site of a lacunar stroke may affect the global accuracy of the CTP study (e.g., lesions outside the coverage volume of CTP, including infratentorial strokes). Some studies limited the evaluation to supratentorial strokes [13], but other studies considered infratentorial strokes, too [17,21]. The detection of infratentorial infarctions can be improved by assessing whole-brain CTP, but it remains a diagnostic challenge, and especially small volume infarctions in the brainstem are likely to be missed [25], being the sensitivity very low [28.1% (95% CI: 12.9; 53.4)]. Globally, the sensitivity for infratentorial infarction <19 mm is 22.9% [95% CI: 9.9; 45.0], even for whole brain CTP [25]. In Zhu [13], 11/29 (37.9%) lacunar strokes were infratentorial; in Eckert 2011 [17], 4/54 of false negative CTP had infratentorial infarction. In Garcia-Esperon [15], most of the lacunar syndromes had a supratentorial infarct on DWI MRI (92 patients, 86.8%), 59 (55.6% of the total) had an isolated subcortical lesion, and 33 (31.1%) had a lesion with cortical involvement. In Tan [8], 31 (17%) patients had infarcts located in the lentiform nucleus 9 (29%), corona radiata 9 (29%), posterior limb of the internal capsule 6 (19%), thalamus 5 (16%), corpus callosum 1 (3%), and brainstem 1 (3%); 55% of subjects had lacunar mechanism. In Das [16], the distribution of LI was lenticulostriate 34 (57.6%), thalamus/posterior internal capsule 11 (18.6%); and pons 14 (23.7%). Perfusion abnormality was seen in 36/59 (61%) LI. In Cao [14], 32 patients had MRI-confirmed lacunar stroke within CTP coverage (18 in the striatum, 10 in the thalamus, 4 in corona radiata). 

The location of RSSI (internal/external capsule/lentiform nucleus vs centrum semiovale vs thalamus) may affect the sensitivity of CTP in identifying them also in the supratentorial areas and within the coverage area of the CTP slab. Valdés Hernández et al. [31] reported the volumes of RSSI in various locations in the acute phase: internal capsule/lentiform nucleus 1.23 mL [0.99–1.77], centrum semiovale 1.32 mL [1.05–1.86], thalamus 0.51 mL [0.32–0.81], brainstem 0.72 mL [0.56–0.98], optical radiation 1.60 mL [1.51–1.69]. It is possible that thalamic lesions may often be under the resolution of CTP because of the relatively low volume in comparison with other locations. Another issue is the relation between RSSI location and the rate of identified perfusion abnormalities. In an MRI study [32] the infarction localization is significantly associated with the frequency of a perfusion deficit: 10/13 (76.9%) in the basal ganglia, 15/27 (55.6%) in the internal capsule, 20/27 (74.1%) in the thalamus, 11/34 (32.4%) in the corona radiata, and 3/10 (30.0%) in the brainstem (*p* = 0.003). 

RSSI, as currently defined, does not have a lower-size boundary. Therefore, this label may also include very small, punctate DWI hyperintense lesions in the subcortical white matter. In Tao et a. [33], a consecutive retrospective cohort of patients with DWI-diagnosed RSSI was separated into acute subcortical microinfarctions (diameter < 5 mm) versus the larger RSSI (diameter 5–20 mm). They found that 23/584 (3.9%) of ischemic lesions were in the first group and located in the basal ganglia (11/23), followed by the thalamus (5/23) and centrum semiovale (4/23).

The year of stroke ranged from 2004 [9] to 2021 [13], and this issue may directly affect the CTP technology (scanner and post-processing software), but this issue cannot be more deeply addressed because of the general heterogeneity of the technical equipment in individual studies. 

Not all studies reported the rate of rtPA administration in the described cohort and the rate of “negative” DWI in patients receiving rtPA, but it is possible that this issue should be taken into account, in particular in old studies with late MRI assessment from the symptom’s onset and patients with long-lasting neurological deficit. Moreover, it is coherent with the previously detailed size issue [33]. Akhtar et al. [20] reported the administration of rtPA among RSSI patients as follows: 213 did not receive rtPA, and 45 received it. In Eckert [17], 84 patients were included <3 h and 23 >3 h; rtPA was administered in 51 patients, 30 ≤ 3 h and 11 > 3 h. Interestingly, Rudilosso [23] reported false negatives in 37.5% of treated patients vs 18.8% of not treated patients. In Zhu [13], 4 (13.8%) lacunar infarcts received rtPA before MRI. Old studies reported the disappearance of MRI perfusion and diffusion abnormalities in lacunar infarctions after thrombolysis [34].

The high sensitivity of DWI-MRI allows to detect very small infarcts (1–2 mm in diameter) [35], but the blooming effects may lead to an overestimation of the maximum diameter of acute DWI in comparison with the true infarct size [36]. Even patients with a transient neurological deficit (TIA) demonstrate DWI lesions at a rate of 1 in 6 to 2 in 3. Symptom duration, speech or motor symptoms, and etiology seem to correlate with the rate of DWI positivity [37]. However, the odds for detectable lesions on DWI seem to decline in the subacute stage with advancing latency between symptom onset and MR imaging [38]. Interestingly, the ADC decrease in the ischemic lesion (if present) was less pronounced in patients with complete recovery in less than 24 h compared to patients with completed strokes [22,39,40,41]. It is virtually unknown if rtPA administration in patients with long-lasting deficits may affect the DWI-ADC appearance of the lesion. In stroke patients, the DWI detection rate of ischemic lesions is >95% [26,27,42,43,44]. Nevertheless, there are reports on DWI-negative strokes [45,46] illustrating that the diagnosis of acute ischemic stroke cannot be excluded solely on the base of DWI without visible lesions. 

In CTP studies, the timing of MRI in follow-up ranged from 2 [7,14,16] to 15 days [8]. This issue may have a great impact on the diagnosis of lacunar stroke if defined by DWI-MRI hyperintensity. In a retrospective analysis, [47] of acute stroke patients divided in DWI positive and DWI negative on MRI performed within 72 h from hospital admission, 15/33 (45.4%) of negative DWI patients had a lacunar syndrome according to Oxfordshire Community Stroke Project Classification (OCSPC) and 20/33 (60.6%) an SVD mechanism. It is possible to propose the hypothesis that using a different MRI technology with a lower sensitivity and extending the timing of scanning from admission can affect the diagnostic accuracy based on the follow-up comparator. The lack of DWI hyperintensity in the initial MRI of acute stroke was reported in 9.5% at 3T [47], 5.8% [45], and 25.6% [48] at lower field strengths. Considering that the clinical symptoms and signs poorly discriminate the size of subcortical infarcts and even the smallest lesions could cause overt neurological symptoms [33], it is likely that some of the DWI-negative stroke patients with long-lasting deficit may be included in this group in particular, if the MRI scanning time was not early.

The selected studies, as reported in Table 2 and Table 3, had a marked heterogeneity in several technical issues about the scanner, and each of them may affect the sensitivity and specificity of the findings. For example, the rows number ranges from 16 [14] to 320 [21], suggesting a completely different reliability of the findings in small subcortical infarctions, the latter a whole brain perfusion technique, covering from the infratentorial region to the vertex and including the entire subcortical white matter. The rows number has a direct impact on volume coverage with a single slab (see Table 3), ranging from 2.8 cm with 64 rows, 10 cm with 128 rows, and up to the entire brain with 256 and 320 rows.

The different technology (16 rows scanner) may have affected the finding from Lin [9] that small hyperacute infarcts, many of the lacunar types, were poorly discernible on CTP, which detected only 2 (15.4%) of the 13 that were within its coverage volume. Eckert [17], using a 40 rows scanner, reported that follow-up MRI detected brain infarcts in 23/54 patients with normal CTP, the majority being lacunar infarcts (16) within the CTP perfusion coverage. Hana [18], with a 64 rows scanner, reported a sensitivity of 0% for lacunar infarction. Campbell [25], with a 16 rows scanner, reported that the main reason for non-diagnostic CTP was lacunar infarction (28 pts—10%), followed by infarct outside slab coverage (21 pts—8%), technical failure (4 pts—1%) and reperfusion (2 pts—0.7%).

In Hana (64 rows scanner) [18], the toggling table CT technique was used to extend the volume coverage of the brain [49]. This technique was found to be useful as an initial imaging method in acute ischemic stroke in order to extend the volume coverage, although it had low sensitivity for detecting small acute infarctions. In particular, this technique provided higher lesion detection than 20-mm-coverage perfusion CT [50]. As a confirmation of these pitfalls, a 0% sensitivity was reported for lacunar infarction by the authors. The same consideration also applies to the shuttle mode used by Tan [8] for the same purpose, i.e., to extend the volume coverage of CTP [51], reducing the temporal resolution [28].

Another source of variability is the use of different (commercially available and vendor machines) software for post-processing CTP source images and generating perfusion maps. The majority of studies declared that visual inspection of the maps was used instead of the automatic output of the software, finding a higher sensitivity: 42% vs. 6% in Garcia-Esperon [15]. In general, the main problem is the relatively low sensitivity and specificity of CTP for lacunar strokes/RSSI; indeed, no false positive perfusion images were rated [14,17,18]. Table 3 summarizes the different perfusion maps assessed in the available studies, and Table 4 are reported the sensitivity, specificity, PPV, and NPV values provided by the individual studies. In Cao [14] MTT map showed 56% sensitivity, Tmax 25% (*p* < 0.001), CBV 9% (*p* = 0.021), and CBF 44% (*p* < 0.001). Using all maps gained 56% sensitivity with a specificity of 100%, PPV of 100%, and NPV of 68%. The better sensitivity of the MTT map was also found by Tan [8], with 12/31 (39%) lacunar infarcts patients having a perfusion deficit compared with those with any cortical infarction (120/142, 67%). In Rudilosso [7], with a 128 rows scanner and 98 mm of vertical coverage in CTP acquisition, sensitivity and PPV of CTP for lacunar stroke were higher than non-contrast CT (63% vs. 19%). Conversely, the specificity was low (20%) and influenced by low lacunar stroke prevalence. The most informative map for the identification of ischemic lesions was TTD. In Benson [6], TTP were the maps with the highest sensitivity (49%); specificity was high regardless of the map evaluated (all > 97%). In Garcia-Esperon [15], the automated core-penumbra CTP maps had a decreasing sensitivity for cortical lesions (56.1%), posterior infarcts (25%), and subcortical infarcts (5.9%) but with high specificity (between 87% and 99%). CBV and CBF maps showed low sensitivity for all regions (<20%) with high specificity (>95%). Conversely, MTT and DT maps performed with a low sensitivity for posterior lesions (<30%), moderate for subcortical (between 30% and 40%), and good for cortical lesions (DT 60.6% versus 51.5% for MTT). Assessing together all the CTP maps dramatically increased the detection of subcortical lesions compared to automated core-penumbra maps (42.4% versus 5.9%). Specificity values were over 80% for both modalities for all the regions. In Farooqui [21], increased TTP was observed in 23 (47%) patients, and it was a predictor of ND on multivariate analysis. In general, although the technical differences in the individual studies, time-based maps were found to be more sensitive than flow-based maps. These latter ones performed better in lesions larger than 20 mm caused by multiple perforating branches occlusion [13].

The CTP technique is performed with significant variability between different institutions, and the CTP parameters are affected by the generation of CT scanners, processing software, and optimization in a single institution in the setting of LVO-related stroke [51]. Another source of variability, besides image acquisition and post-processing techniques, is provided by the intrinsic features of cerebral perfusion [51]. In fact, both physiologically and in acute ischemia, brain perfusion is not the same in all areas because of the neuronal activity of different regions, leading to different perfusion patterns in patients with similar occlusion sites and times from stroke onset to imaging acquisition. The efficiency of intrinsic compensatory mechanisms (collateral blood supply, vasodilation, and oxygen extraction at capillary levels) influences this variability in perfusion.

Finally, there are other limitations that increase the heterogeneity and the transferability of the presented data. If a mild variability emerges in the timing of CTP study from symptom onset because most patients were enrolled within routine hyperacute stroke pathways management, the eligibility for CTP was not the same in all studies. In Akhtar [20], 92 + 38 patients were studied by CTP, and the main reason for not doing CTP was a clinical diagnosis of small vessel (lacunar) stroke. In Campbell [10], the rate of patients undergoing CTP increased during the enrollment period. 

A potential issue affecting the sensitivity of CTP for small subcortical perfusion abnormalities is the coexistent leukoaraiosis, which in some grades is expected in patients with SVD-related stroke [7,52,53]. Consistent with the pathogenesis of leukoaraiosis, hypoperfusion detected with CTP was found to be associated with white matter disease severity [54,55]. It has been reported that leukoaraiosis mat confounds the CTP estimation of the final infarction in patients undergoing mechanical thrombectomy [56], but this issue has not been addressed for patients with non-territorial infarctions. The effect of leukoaraiosis has been compensated by the restriction of the core to voxels with both low rCBF and delayed TTP and by automatic detection and removal from the analysis by a Housfield unit threshold [57].

It is possible that a perfusion evaluation distinguishing the penumbra from the core is not possible for RSSI in the same way it is performed for small and large territorial infarctions. The resolution of CTP maps does not allow this task, and it is well described in Garcia-Esperon [15], where, in the lacunar group, the median core volume was 0 [0–0.9] mL [IQR], the median penumbra volume 0.4 [0–2.9] mL [IQR] and the median hypoperfused lesion volume 0.7 [0–4.6] mL [IQR]. These measures correspond to a median volume in DWI of 1.9 [0.8–5.5] mL [IQR], so DWI hyperintensity size is greater than the size of hypoperfused tissue in CTP. Indeed, CTP maps have a relatively low spatial resolution, and small infarcts can be omitted even when they are included in the covered volume. Areas of chronic infarction are generally obvious on NCCT; however, in perfusion, they can be confusing. Most of the tissues with chronic infarction show a low but persistent degree of metabolism and decreased but measurable perfusion parameters [58]. 

Moreover, gray and white matter do not have the same physiological perfusion features and thresholds. Indeed, gray matter presents higher values of CBF and CBV but lower values of MTT and venous drainage time (VDT), which is in direct relation to its higher energy consumption [58]. Conversely, time-dependent maps (MTT and DT) show higher values in white matter than in the gray matter. It is critical that the contralateral regions of interest used for normalization maintain the same gray matter/white matter ratio as the ipsilateral ischemic region. In fact, CBV and CBF have different baseline values in gray and white matter (CBV: gray matter CBV 4 mL/100 g/min, white matter CBV 2 mL/100 g/min; CBF: gray matter CBF 40–60 mL/100 g/min, white matter CBF 20–30 mL/100 g/min), [59].

CTP appears as an imperfect technique if the aim is to identify an RSSI in comparison to the more sensitive DWI-MRI, which is, in turn, a less imperfect technique but far to be 100% performant. The very low signal-to-noise ratio and the correspondent low contrast in CTP source images and perfusion maps may prevent to identify of small subcortical infarcts, but, as previously detailed, the potential reasons for not detecting a perfusion deficit are several, and in some cases, a perfusion deficit is not present at all. CTP and PWI-MRI studies tell us a different story about the pathophysiology of lacunar infarctions and the underlying mechanisms of tissue damage, growth, and long-term evolution and support the existence of a heterogeneous perfusion pattern in RSSI, as discussed in Zedde et al. [60]. 

## 5. Conclusions

Evolving techniques may complete the missing pieces of the unknown about the dynamics of RSSI, although the main aim of CTP in acute stroke pathway management is to identify viable tissue in large and medium vessel occlusions in order to select patients amenable to endovascular treatment. The main conclusion is that the heterogeneity and limitations of CTP techniques are associated with a previously neglected heterogeneity in RSSI perfusion and evolution. Now, RSSI cannot be reliably diagnosed using CTP because the rate of false negatives is still consistent, but it may support the clinical diagnosis in the early phase when a perfusion deficit is seen. Larger, technically homogeneous, and prospective studies are needed to better define the role of CTP in the diagnosis of RSSI. 

## Figures and Tables

**Figure 1 diagnostics-13-01564-f001:**
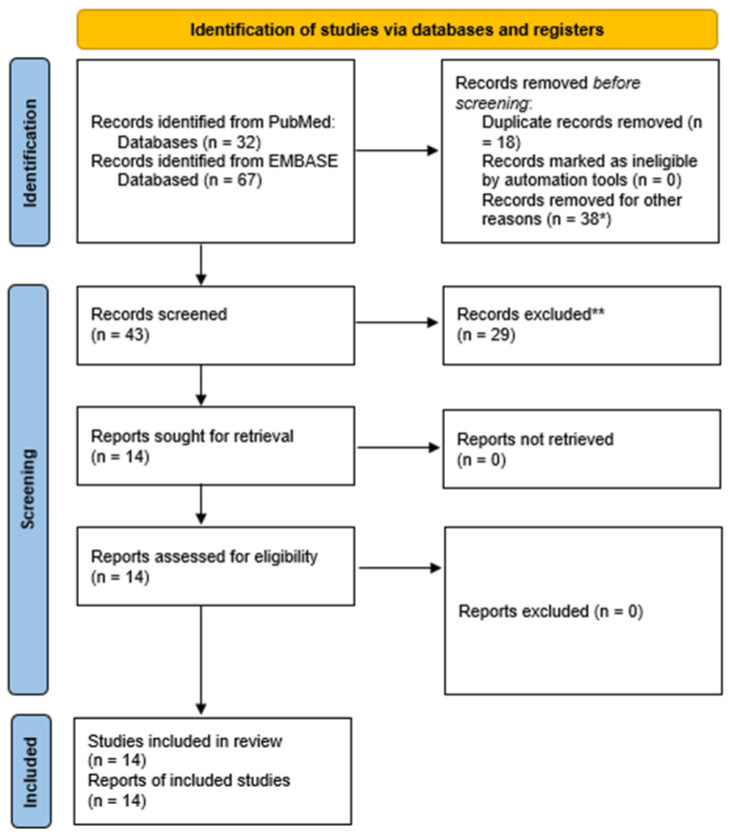
PRISMA 2020 flow diagram. *, 10 case reports, 10 papers about nonlacunar stroke, seven studies not using CTP, four papers not pertinent to stroke, two replies, two pictorial or narrative reviews, two conference abstracts, one animal model. **. There are nine studies without data con CTP in lacunar stroke, six studies on nonlacunar stroke, five narrative reviews, three papers focused on technical protocol, two conference abstracts, two systematic reviews not focused on lacunar stroke, 1 study protocol, one letter without new data.

**Table 1 diagnostics-13-01564-t001:** Main methodological data and definitions from the selected papers.

Reference	Country	Study Design	Study Period	Population	Lacunar Stroke Definition	Neuroimaging Comparator
Zhu 2022 [13]	China	Monocenter retrospective cohort	January–December 2021	599 CTP examinations in patients with acute stroke symptoms within 12 h and MRI within 7 days of symptom onset	Intracerebral lesions < 20 mm in diameter and fewer than 3 slices or a lesion within the pontine parenchyma on DWI-MRI	MRI
Tan 2016 [8]	Australia	Monocenter retrospective cohort	April 2009–March 2013	182 patients with ischemic stroke	Single subcortical infarct ≤ 20 mm on follow-up MRI	MRI (median time 15 days)
Cao 2016 [14]	Australia	Retrospective cohort	2009–2013	63 patients (32 with RSSI and 30 without lesions on DWI)	Ovoid DWI hyperintensities with maximum diameter ≤ 25 mm in the territory of small penetrating arteries (thalamus, striatum, or corona radiata)	MRI (within 48 h)
García-Esperón 2021 [15]	Australia	Retrospective analysis	January 2010–June 2018	712 DWI-confirmed stroke patients	Combination of clinical lacunar syndromes, hyperintensity on DWI-MRI in a penetrating vessel territory and no visible vessel occlusion in that region	DWI-MRI
Rudilosso 2015 [7]	Spain	Retrospective analysis	January 2009–December 2012	33 patients with lacunar syndrome	Infarct volume < 1.767 cm^3^ (the volume of a sphere with a diameter 1.5 cm) on DWI	MRI; median delay 26 h (IQR 18–43 h)
Das, 2015 [16]	UK	Retrospective cohort	March 2011–February 2013	88 patients with lacunar syndrome RSSI: 59/88 (67%).	Solitary, subcortical lesions, ≤ 20 mm in greatest diameter	MRI (median time 2d, IQR 1–4.25 d)
Benson 2016 [6]	USA	Retrospective cohort	January 2006–July 2011	113 patients: 37 (32.7%) lacunar	Infarcts < 20 mm in maximum diameter on DWI (either cortical or subcortical)	MRI within 7 days
Lin 2009 [9]	Germany	Retrospective cohort	January 2004–May 2008	65 consecutive stroke patients < 3 h and receiving NCCT and CTP and follow-up DWI < 7 days	DWI < 15 mm	DWI-MRI within 7 days
Eckert 2011 [17]	Germany	Retrospective cohort	April 2007–April 2008	107 stroke patients within 6 h	Not stated	MRI within 2 to 5 days CT if contraindication to MRI
Campbell 2013 [10]	Australia	Retrospective cohort	January 2009–September 2011	277/475 (58%) ischemic stroke patients < 9 h/wake-up onset studied by CTP	Not stated	CT or MRI
Hana 2014 [18]	Japan	Retrospective cohort	October 2008–October 2011	87 patients with ischemic stroke with CTP at the admission and DWI-MRI post hospitalization	Not stated	DWI-MRI
Rudilosso 2019 [19]	Spain	Retrospective cohort	May 2011–September 2016	Acute stroke patients undergoing CTA + CTP	RSSI according to the STRIVE definition (a symptomatic hyperintensity in the territory of one perforating arteriole measuring < 20 mm in its maximum diameter in the axial plane on DWI)	DWI-MRI
Akhtar 2020 [20]	Qatar	Retrospective cohort	January 2014–April 2017	506 acute stroke patients	Not stated	NA
Farooqui 2022 [21]	USA	Retrospective cohort	March 2015–June 2016	Patients with lacunar stroke who had magnetic resonance imaging and CTP performed	Not stated	DWI-MRI

CTP: CT Perfusion; DWI: diffusion-weighted imaging; MRI: Magnetic Resonance Imaging; NCCT: noncontrast CT.

**Table 2 diagnostics-13-01564-t002:** Main demographic and neuroimaging findings.

Reference	Lacunar Stroke Patients (N)	Women (%)	Mean Age (years)	CTP Technical Notes	CTP Findings
Zhu 2022 [13]	29	37.9	66.9 ± 9.4	**Scanner**: 64-section multidetector scanner (Revolution Frontier, GE Healthcare, Siemens) **Post-processing software**: d fast-processing of ischemic stroke (F-STROKE) software [22] **Perfusion maps:** TTP, CBF, CBV, MTT	5/18 (27.8%) in the lacunar stroke group had perfusion deficits in the supratentorial region
Tan 2016 [8]	31	44	66.4 ± 15.3	**Scanner:** General Electric 750HD 64-slice CT scanner or a Philips 128-slice CT scanner **Post-processing software**: Advantage Windows (GE Medical Systems) and Extended Brilliance Workspace (Philips Healthcare, Best, Netherlands) **Perfusion maps**: MTT	12/31 (39%) lacunar stroke patients had a perfusion deficit compared with those with any cortical infarction (120/142, 67%)
Cao 2016 [14]	32	32	70 ([IQR] 56–79)	**Scanner**: Somatom 16-slice scanner, Siemens, Erlangen, Germany) **Post-processing software**: RAPID iSchemicView (Menlo Park, CA, USA) **Perfusion maps**: CBF, CBV, MTT, and Tmax	The sensitivity of CTP vs. DWI-MRI was 56% for MTT, and significantly lower for Tmax (25%, *p* < 0.001), CBV (9%, *p* = 0.021) and CBF (44%, *p* < 0.001). Using all four maps gained 56% sensitivity, 100% specificity, 100% PPV and 68% NPV.
García-Esperón 2021 [15]	59/106 (55.7%) (59 RSSI, 33 cortical and 14 infratentorial strokes)	NA	70 (59–79)	**Scanner**: Aquilion 320-slice CT scanner (Toshiba) **Post-processing software**: MIStar (Apollo Medical Imaging Technology, Melbourne, Australia) **Perfusion maps**: CBF, CBV, MTT, DT	42% SE, 80% SP for RSSI. Visual inspection of CTP maps had higher sensitivity than the automated method (42% vs. 6%).
Rudilosso 2015 [7]	16	37.5	62.6	**Scanner**: Somatom definition Flash 128-slices dual-source CT system **Post-processing software**: visual assessment by using CT Neuro Perfusion Syngo.via (Siemens Healthcare GmbH); core/penumbra threshold analysis by using MIStar (Apollo Medical Imaging Technology, Melbourne, Australia) **Perfusion maps**: CBF, CBV, MTT, Tmax, TTP, TTD and MIP	The sensitivity and PPV of CTP were higher than NCCT (63% vs. 19%). The sensitivity of CTP was higher for supratentorial than for infratentorial (65% vs. 16%). Specificity was low (20%). TTD was the most informative map.
Das, 2015 [16]	59 (14/59 pontine)	NA	72	**Scanner**: 64-section CT scanner **Postprocessing software**: CT Perfusion 4, GE Healthcare **Perfusion maps**: CBV, CBF and MTT	36/59 (61%) with a concordant abnormal findings between DWI-MRI and CTP
Benson 2016 [6]	37 (all supratentorial)	53.1	59.8 ± 16.9	**Scanner**: 64-section CT scanner **Post-processing software**: Vitrea workstation (Vital Imaged, Minnetonka, Minnesota) **Perfusion maps**: TTP, MTT, CBV and CBF	Sensitivity was highest for TTP (49%), and lowest for NCCT (3%). Specigicity mantained high with any map (all > 97%).
Lin 2009 [9]	13/65	NA	NA	**Scanner**: 16-slice scanners (Siemens Sensation, Siemens AG, Erlangen, Germany) **Postprocessing software**: Siemens Syngo Neuro Perfusion CT, Siemens AG **Perfusion maps**: CBF, CBV, TTP	CTP detected 2/13 (15.4%) LI that were within its coverage volume
Eckert 2011 [17]	54 with no occlusion on CTA	NA	68.4	**Scanner**: 40-row multidetector CT scan, Brilliance 40 (Philips, Eindhoven, The Netherlands) **Post-processing software**: Philips Medical System, Best, The Netherlands **Perfusion maps**: MTT, CBF, CBV and TTP	23/54 (42.59%) patients had normal MMCT and positive follow-up MRI 16/23 (69.56%) were lacunar infarcts within the CTP perfusion maps, 4 were infratentorial infarcts and 3 were territorial infarcts beyond the perfusion maps.
Campbell 2013 [10]	NA	NA		**Scanner**: Siemens Somatom 16-slice multidetector scanner (Siemens, Erlangen, Germany) **Postprocessing software**: Siemens Syngo NeuroPCT, Siemens **Perfusion maps**: TTP, CBF, CBV Visual assessment of unthresholded TTP and CBV/CBF maps	Non-diagnostic CTP was due to lacunar infarction (28 (10%)), infarct outside slab coverage (21 (8%)), technical failure (4 (1%)) and reperfusion (2 (0.7%)).
Hana 2014 [18]	NA	NA		**Scanner**: 64 row detector CT scanner (LightSpeed VCT XT; GE) **Post-processing software**: CTP software developed by GE Health care **Perfusion maps**: MTT, CBV, CBF	55/87 CTP deficit with highr sensitivity in infarcts < 3 cm; sensitivity 0% for lacunar infarction
Rudilosso 2019 [19]	74 RSSI	40	66.3 (12.1)	**Scanner**: Somatom Definition Flash 128-section dual-source CT system (Siemens, Erlangen, Germany) **Postprocessing software**: Syngo CT Neuro Perfusion VA20 (Siemens) **Perfusion maps**: CBF, CBV, TTP and TTD	On visual inspection, hypoperfusion on CTP was identified in 51 patients (76%), perfusion was normal in 12 patients (18%) and hyperperfusion was observed in the remaining 4 patients (6%).
Akhtar 2020 [20]	130	NA	NA	**Scanner**: NA **Postprocessing software**: NA **Perfusion maps**: NA	The main reason for not doing CTP was a clinical diagnosis of small vessel (lacunar) stroke.
Farooqui 2022 [21]	49 (19 ND + 30 no ND)	25 (10 + 15)/49 51.02%	65 ± 13 vs. 62 ± 15	**Scanner**: 320–detector row scanner (Toshiba Aquilion One; Toshiba Medical Imaging, Tokyo Japan) **Postprocessing software**: Vitrea; Vital Images Minnetonka, Minnesota, USA **Perfusion maps**: CBF, CBV, MTT, TTP	63% of patients had stroke diameters < 10 mm. ND patients have a trend for increased TTP in the stroke area compared to the no ND patients [12 (63%) vs. 11 (37%), *p* = 0.07]. 8 patients had a change in CBV without difference between the two groups. Increased TTP was a predictor of ND on multivariate analysis. MTT was not assessed, considering TTP more sensitive.

CBF: cerebral blood flow; CBV: cerebral blood volume; CTP: CT perfusion; DT: delay time; DWI: diffusion-weighted imaging; MIP: maximum intensity projection; MTT: mean transient time; ND: neurological deterioration; NPV: negative predictive value; NCCT: noncontrast CT; PPV: positive predictive value; RSSI: recent small subcortical infarct; SE: sensitivity; SP: specificity; Tmax: time to maximum; TTD: time to drain; TTP: time to peak.

**Table 3 diagnostics-13-01564-t003:** Perfusion maps were examined in the selected studies.

Reference	CBF	CBV	MTT	DT ^1^	TMax	TTP	TTD	Vertical Coverage (mm)
Zhu 2022 [13]	Y	Y	Y			Y		NR
Tan 2016 [8]			Y					80 ^2^
Cao 2016 [14]	Y	Y	Y			Y		
García-Esperón 2021 [15]	Y	Y	Y	Y				160
Rudilosso 2015 [7]	Y	Y	Y		Y	Y	Y	98
Das, 2015 [16]	Y	Y	Y					NR
Benson 2016 [6]	Y	Y	Y			Y		80 ^4^
Lin 2009 [9]	Y	Y				Y		24
Eckert 2011 [17]	Y	Y	Y			Y		40
Campbell 2013 [10]	Y	Y				Y		24 ^4^
Hana 2014 [18]	Y	Y	Y					80 ^3^
Rudilosso 2019 [19]	Y	Y				Y	Y	98
Akhtar 2020 [20]	NA	NA	NA	NA	NA	NA	NA	NR
Farooqui 2022 [21]	Y	Y	Y			Y		160

^1^ DT maps are calculated very similarly to Tmax (time to maximum) and are used to define the perfusion lesion, with slightly better accuracy; ^2^ shuttle mode; ^3^ toggling table CT technique [26,27]; ^4^ Two slabs; NR: not reported; NA: not assessed. CBF: cerebral blood flow; CBV: cerebral blood volume; CTP: CT perfusion; DT: delay time; MTT: mean transient time; Tmax: time to maximum; TTD: time to drain; TTP: time to peak. Y: yes.

**Table 4 diagnostics-13-01564-t004:** Sensitivity and specificity data of CTP for RSSI.

Reference	Sensitivity	Specificity	PPV	NPV
García-Esperón 2021 [15]	CTP core/pen: 5.9%	CTP core/pen: 93.6%	CTP core/pen: 53.8%	CTP core/pen: 44.2%
	CBV 2.5%	CBV 97.9%	CBV 60%	CBV 44.4%
	CBF 5.1%	CBF 97.9%	CBF 75%	CBF 45.1%
	MTT 39.8%	MTT 90.4%	MTT 83.9%	MTT 54.5%
	DT 33.9%	DT 91.5%	DT 83.9%	DT 52.4%
	Any positive map: 42.4%	Any positive map: 89.4%	Any positive map 83.3%	Any positive map: 55.3%
Rudilosso 2015 [7]	62.5%	20%		
Das, 2015 [16]	9.3–42.5%	91.9–95.3%		
Benson 2016 [6]	All sites: 18.9% (CBV), 48.7% (TTP) CSWM 65.2% PWM: 12.5–37.5% BGT: 0%	All sites: 97.4% (CBF-TTP), 98.7% (CBV-MTT)	81.8% (CBF), 92.9% (MTT)	71.4% (CBV), 80% (TTP)
Lin 2009 [9]	15.4%	100%	100%	82.5%
Eckert 2011 [17]	0%	100%	0%	100%
Hana 2014 [18]	0%	100%	0%	100%
Cao 2016 [14]	MTT 56% Tmax 25% CBV 9% CBF 44% All maps 56%	All maps 100%	All maps 100%	All maps: 68%

CBF: cerebral blood flow; CBV: cerebral blood volume; CTP: CT perfusion; DT: delay time; MTT: mean transient time; PPV: positive predictive value; Tmax: time to maximum; TTP: time to peak.

## Data Availability

Not applicable.

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
