# Peer review of "CT Perfusion in Lacunar Stroke: A Systematic Review"

_diagnostics, 2023, doi:10.3390/diagnostics13091564_

Round 1

Reviewer 1 Report

The authors present a systematic review of scientific publications on CT perfusion imaging in lacunar stroke published between 2000 and 2022.

They concluded that CTP imaging cannot be used to exclude lacunar infarct in case of absence of a perfusion deficit.

Their paper is based on an extensive literature research.

Unfortunately, the discussion part seems to be not stringent enough to me . It is difficult to follow a clear cut message. I have the feeling this part could be improved by shortening.

As it is general knowledge from everyday clinical routine that CTP imaging often is not helpful in lacunar infarcts to guide whether IVT in the extended time window can be applied safely, I have doubts that the review is of great interest to the scientific community.

Additionally, the conclusion made by the authors has low implication for clinical decision making for my feeling.

Author Response

The authors present a systematic review of scientific publications on CT perfusion imaging in lacunar stroke published between 2000 and 2022.

They concluded that CTP imaging cannot be used to exclude lacunar infarct in case of absence of a perfusion deficit.

Their paper is based on an extensive literature research.

First of all, we would thank the reviewer for the comments on our paper. The focus of our systematic review is to highlight that a perfusion deficit is present in about 50% of supratentorial lacunar infarcts by using CTP but the lack of identification of such a deficit has several reasons and one of them is that sometimes there is not a perfusion deficit at all in the pathophysiology of pure SVD-related lacunar stroke (i.e. the stroke due to the occlusion of a single perforating branch). Therefore, not only technical issue related to CTP play a role in the low sensitivity for lacunar stroke in the hyperacute phase.

Thanks again for noting the extensive literature screening. 

Unfortunately, the discussion part seems to be not stringent enough to me . It is difficult to follow a clear cut message. I have the feeling this part could be improved by shortening.

We agree with the reviewer and, accordingly also with the suggestion of another reviewer we largely shorthened the discussion, skipping the pathophysiological issues for another paper (it was suggested to write a narrative review on this topic). In the revised version of the paper it can be recognized the change, mainly in the introduction and in the discussion sections.

As it is general knowledge from everyday clinical routine that CTP imaging often is not helpful in lacunar infarcts to guide whether IVT in the extended time window can be applied safely, I have doubts that the review is of great interest to the scientific community.

We fully understand the reviewer's comment, but we respectfully started from a different point of view. The management setting of acute stroke patients is predominantly based on CT imaging rather than MRI, even in the late window. Moreover, even the latter, although more sensitive in particular in the lacunar stroke, has different limitations and not always such a good sensitivity in the hyperacute phase, as detailed in the discussion, in particular in the involvement of the subtentorial regions. That said, the absence of a perfusion deficit in CTP in the late window in a lacunar stroke currently determines the abstention from the administration of fibrinolytic therapy and not necessarily this does not change the patient's prognosis. Although the presence of a perfusion deficit is associated with a greater risk of neurological deterioration in patients with RSSI, this information derives from a limited series and, as detailed in the discussion, the sources of variability that limit the interpretation are multiple. In CT-based settings, the diagnosis of lacunar stroke with CTP in the late window is unfortunately a consistent clinical problem and the better understanding of the physiopathology underlying the neuroimaging findings helps to frame the latter and to make more motivated and aware therapeutic choices.

Additionally, the conclusion made by the authors has low implication for clinical decision making for my feeling.

Again, we understand the point of view raised by the reviewer. Nevertheless, we think that the fully understanding of a phenomenon in its pathophysiology is needed to correctly use and interpret the available diagnostic techniques and ischemic stroke is an heterogeneous disease, not only in large vessels but also in small vessels. The analysis of the limitation of the actual knowledge is the mandatory step to improve it, so our goal has been perfectly reached and if even one vascular neurologist will think about lacunar stroke in a more complex and dynamic way than routinely done in general, our literature review efforts deserved to be made. In a time of rapid decisions and reduction of reasoning on complexity, the message that strokes from SVD have a pathophysiology that is not simple and cannot be trivialized and that this determines the findings on neuroimaging, i.e. an index to asking questions, is the prerequisite for a better treatment.  

Reviewer 2 Report

The authors conducted a systematic review of all studies regarding the sensitivity and specificity of CT perfusion to detect a lacunar stroke during the acute phase. 14 studies that covered a large period (from 2004-2022) were included. Qualitative assessment of studies precluded any further metanalysis due to high heterogeneity. The authors concluded that there were two sources of heterogeneity: one due to the technical characteristics of CT perfusion and one due to the heterogeneity of the pathophysiological mechanism and the definition of lacunar strokes. 

The authors have conducted a thorough research of the bibliography and should be praised for this. The main problem to my point of view is the nature of the manuscript which is both a systematic review and a narrative review regarding the issue. Since they have already conducted a huge effort, I suggest that in fact the authors could provide two different papers with equal validity: one systematic review that will summarize more consisely and briefly the conclusions of their assessment and data extraction. The introduction and the discussion should be limited accordingly. For example, paragraph 2 from introduction and rows 210-277 that are addressing the pathophysiological hypothesis of subcortical infarcts could be omitted in this case. 

Finally, there seems to be a great heterogeneity which goes in parallel to the evolution of CT perfusion technique per se, since 2004: the use of newer generation 256 + CT scanner that can provide whole brain perfusion imaging and also the refinement and optimization of thresholds.  It seems that across different studies visual inspection yields higher sensitivity and specificity for diagnosis compared to thresholds maps , particularly those produced by automatic post processing softwares. Moreover, there is a still unresolved problem regarding different rCBF thresholds in grey and white matter that is reflected in the use of CTP for diagnosis of small subcortical infarcts. 

There was an explosion in the use of CTP during the last decade driven by its use in major RCTs regarding mechanical thrombectomy. However operational criteria and automated software used in these trials were optimized for patients with LVO and consequently large perfusion deficits, so the method has not been still optimized to capture smaller deficits or other neurovascular pathologies (stroke mimics etc). For example coexisting leukoaraiosis has been a major issue for these studies, because it leads to false positive areas of decreased rCBF (rows 477-483). This was resolved by restriction of core to voxels with both low rCBF and delayed Time to Peak and by automatic detection and removal from the analysis by a Housfield unit threshold (see PMID: 22858726

In general, this paper summarizes exactly these challenges and the need for future research.

Lastly I suggest that the paper could be further profited if reviewed by a native speaker for many syntax errors that may obsure the context in some instances for example 390-393

Author Response

The authors conducted a systematic review of all studies regarding the sensitivity and specificity of CT perfusion to detect a lacunar stroke during the acute phase. 14 studies that covered a large period (from 2004-2022) were included. Qualitative assessment of studies precluded any further metanalysis due to high heterogeneity. The authors concluded that there were two sources of heterogeneity: one due to the technical characteristics of CT perfusion and one due to the heterogeneity of the pathophysiological mechanism and the definition of lacunar strokes. 

We would like to thank the reviewer for the attention dedicated to our paper and describing our aims and strategy.

The authors have conducted a thorough research of the bibliography and should be praised for this. The main problem to my point of view is the nature of the manuscript which is both a systematic review and a narrative review regarding the issue. Since they have already conducted a huge effort, I suggest that in fact the authors could provide two different papers with equal validity: one systematic review that will summarize more consisely and briefly the conclusions of their assessment and data extraction. The introduction and the discussion should be limited accordingly. For example, paragraph 2 from introduction and rows 210-277 that are addressing the pathophysiological hypothesis of subcortical infarcts could be omitted in this case. 

Again, we would like to thank the reviewer for the appreciation shown for our work, which was effectively an extensive analysis of the literature in search of the sources of heterogeneity present, starting from the working hypothesis of the existence of a perfusional impairment in a subset of patients with lacunar stroke. The pathophysiological aspects underlying the neuroimaging findings are important and help to understand and interpret these findings. The reviewer's suggestion to separate the two different aspects, i.e. the pathophysiology of lacunar stroke from the perspective of perfusion and the analysis of literature data in the form of a systematic review, into two different papers, a systematic review and a narrative review, It's a great idea and we very much welcomed it. Therefore, we have modified the paper in accordance with the reviewer's indications and we have moved some sections, and in particular those suggested by the reviewer, into a further document, to which we refer at the end of the paper, which will be completed and submitted separately, enriched with the material already available and not used for the systematic review.

In particular, we omitted rows 210-277 and arranged the references accordingly.

Finally, there seems to be a great heterogeneity which goes in parallel to the evolution of CT perfusion technique per se, since 2004: the use of newer generation 256 + CT scanner that can provide whole brain perfusion imaging and also the refinement and optimization of thresholds.  It seems that across different studies visual inspection yields higher sensitivity and specificity for diagnosis compared to thresholds maps , particularly those produced by automatic post processing softwares. Moreover, there is a still unresolved problem regarding different rCBF thresholds in grey and white matter that is reflected in the use of CTP for diagnosis of small subcortical infarcts. 

Once again, the reviewer captured one of the main messages we aimed to spread about CTP and its heterogeneity. Non all regions of the brain are physiologically equal and not all hypoperfused regions of the brain are equally hypoperfused and imaged.

There was an explosion in the use of CTP during the last decade driven by its use in major RCTs regarding mechanical thrombectomy. However operational criteria and automated software used in these trials were optimized for patients with LVO and consequently large perfusion deficits, so the method has not been still optimized to capture smaller deficits or other neurovascular pathologies (stroke mimics etc). For example coexisting leukoaraiosis has been a major issue for these studies, because it leads to false positive areas of decreased rCBF (rows 477-483). This was resolved by restriction of core to voxels with both low rCBF and delayed Time to Peak and by automatic detection and removal from the analysis by a Housfield unit threshold (see PMID: 22858726).

We skipped rows 477-483 and added in the text the suggested reference (rows 363-366). Moreover, an extensive revision of the sintax was performed, rephrasing some sentences.

In general, this paper summarizes exactly these challenges and the need for future research.

Many thanks again.

Lastly I suggest that the paper could be further profited if reviewed by a native speaker.

We performed a complete revision of the text in order to match the request of the reviewer. All changes were tracked in the submitted draft

Round 2

Reviewer 2 Report

All previous comments were sufficiently addressed in the revised manuscript.